# Distribution and Management of the Invasive *Swietenia macrophylla* King (Meliaceae) at the Foot of a Protected Area in Luzon Island, Philippines

Ericson Esquibel Coracero [1,2] 

1   Department of Forestry, College of Agriculture and Forestry, Batangas State University—The National Engineering University Lobo Campus, Lobo 4229, Batangas, Philippines; ericson.coracero@g.batstate-u.edu.ph; Tel.: +63-963-4900-319

2   School of Graduate Studies, Aurora State College of Technology, Baler 3200, Aurora, Philippines

**Abstract:** Invasive alien plant species (IAPS) pose one of the most significant threats to native biodiversity. *Swietenia macrophylla*, or big leaf mahogany, is among the most threatening invasive plants in the Philippines. This article aimed to formally document the presence of *S. macrophylla* along the edges of Mt. Banahaw de Nagcarlan, a protected area on Luzon Island, Philippines. The study also sought to identify the management strategies being implemented by various government institutions to address big leaf mahogany and other invasive plants. A total of 1591 individuals of *S. macrophylla* were documented in mixed land-use areas and roadsides. These were found to have been introduced by the Department of Environment and Natural Resources in 1991 as a reforestation species. Fortunately, no individuals were observed beyond the buffer zone towards the protected area. The identification of management strategies for big leaf mahogany and other IAPS revealed that there is no established approach specifically addressing the presence of *S. macrophylla* at the site. However, some institutions advocate for the conservation of native plants through tree planting activities and educational campaigns. Furthermore, no collaborative efforts were observed among stakeholders and institutions. The results of this study highlight the urgent need to manage the *S. macrophylla* population. Planning and enforcement of strategies require collaborative efforts among stakeholders to prevent its entry into the protected area and ensure the preservation of native biodiversity.

**Keywords:** big leaf mahogany; distribution map; invasive species management; invasive alien plant species; Mt. Banahaw de Nagcarlan

## 1. Introduction

Invasive alien plant species (IAPS) are considered a global problem that poses threats to many ecosystems, thereby causing severe ecological, economic, and social impacts [1]. In the ecological context, IAPS are infamous for disrupting ecosystem processes, altering composition and structure of native plant communities, and outcompeting native plants, leading to the decline in indigenous plant diversity [2,3]. Meanwhile, on the socio-economic aspect, the presence of IAPS can lead to reduced agricultural productivity, damaged power lines and buildings, and disturbed cultural heritage and human well-being [4–6]. Hence, dictating the importance of implementing control and management strategies to reduce the impacts of invasive plant species on native biodiversity and human well-being.

*Swietenia macrophylla* King (Meliaceae) or big leaf mahogany is one of the most prominent and aggressive invasive plants introduced to the Philippines as a reforestation species in as early as 1911 [7]. Back then, its invasive potential remained hidden until it revealed its ability to suppress the growth of other plants under its canopy due to its allelopathic ability [8]. Efforts has been carried out to document and assess the population and impacts of *S. macrophylla* across the Philippines [7,9], thus, declaring it as an invasive species in the country [10]. Unfortunately, this plant already had established populations in some of the

country's protected areas, such as Mt. Makiling Forest Reserve [11], Ninoy Aquino Parks and Wildlife Center [12], and Rajah Sikatuna Protected Landscape [13], among others.

Despite there being several studies documenting the presence of *S. macrophylla*, there is a scarcity of scientifically accepted information regarding the population and distribution of the species in Mts. Banahaw–San Cristobal Protected Landscape (MBSCPL), including its neighboring zones (e.g., buffer zone). Generally, there are very limited studies about tree species composition and diversity in MBSCPL. Only three studies are publicly available which only focused on biodiversity assessment [14,15], the conservation status of plants [16], and the selection of plus and mother trees [17]. Furthermore, most of these studies were only focused on one side of the mountain located in the province of Quezon. Thus, the portion of the protected area located in Laguna is relatively unexplored as the lack of studies indicate. Nonetheless, local community reports and personal observation of the presence of big leaf mahogany within the area has prompted the author to conduct the study in MBSCPL.

MBSCPL is one of the protected areas under the Republic Act No. 7586 or the National Integrated Protected Areas System Act of 1992 (NIPAS Act of 1992) [18]. It is situated in the provinces of Laguna and Quezon in Southern Luzon, Philippines [19]. This holds unique keystone organisms, such as the endemic *Rafflesia banahaw* [20] and the members of one of the most threatened family, Dipterocarpaceae, including *Shorea contorta* and *Parashorea malaanonan* [16]. As a protected zone, the conservation of these biodiversity resources must be a top priority of the concerned institutions, thus, protecting them from disturbances and damages such as the threats brought by IAPS.

Gathering information about IAPS, such as its spatial distribution and existing management strategies, can provide basis for an improved management. Specifically, this information is the base for the determination of key areas for monitoring and control, development of targeted management strategies, and informing wise decision-making on the management and control of the species to reduce its negative impact [21,22]. In line with this, the author aimed to pioneer the documentation of the invasive big leaf mahogany at the edges of Mt. Banahaw de Nagcarlan, the portion of MBSCPL located in Nagcarlan, Laguna. This study can serve as a tool to address the knowledge gap on the presence of the species in the area by locating the populations of big leaf mahogany, as well as the existing management practices of the government agencies with regard to *S. macrophylla*.

## 2. Materials and Methods

### 2.1. Study Area

The study was conducted from March to May 2023 at the foot barangay of Mt. Banahaw de Nagcarlan, in Laguna, specifically Kanluran Lazaan (Figure 1). This barangay is within the boundary (forest reservation) of Mts. Banahaw–San Cristobal Protected Landscape (MBSCPL). The mountain was declared a protected landscape under the Mts. Banahaw—San Cristobal Protected Landscape (MBSCPL) Act of 2009. The protected landscape covered around 10,900.59 ha of lowland to submontane forest [19]. The study was specifically conducted at the barangay at the foot of the mountain, where agriculture and agroforestry were as dominant as residential areas. The topography in these areas was moderate to steep. MBSCPL generally belonged to climate type II with a sandy clay loam soil [23]. The average temperatures in Nagcarlan, Laguna during the study period were 31 °C during the day and 23 °C at night [24]. It typically receives 152.3 mm of rainfall and experienced about 8 rainy days in the month, and the humidity level was usually around 240%.

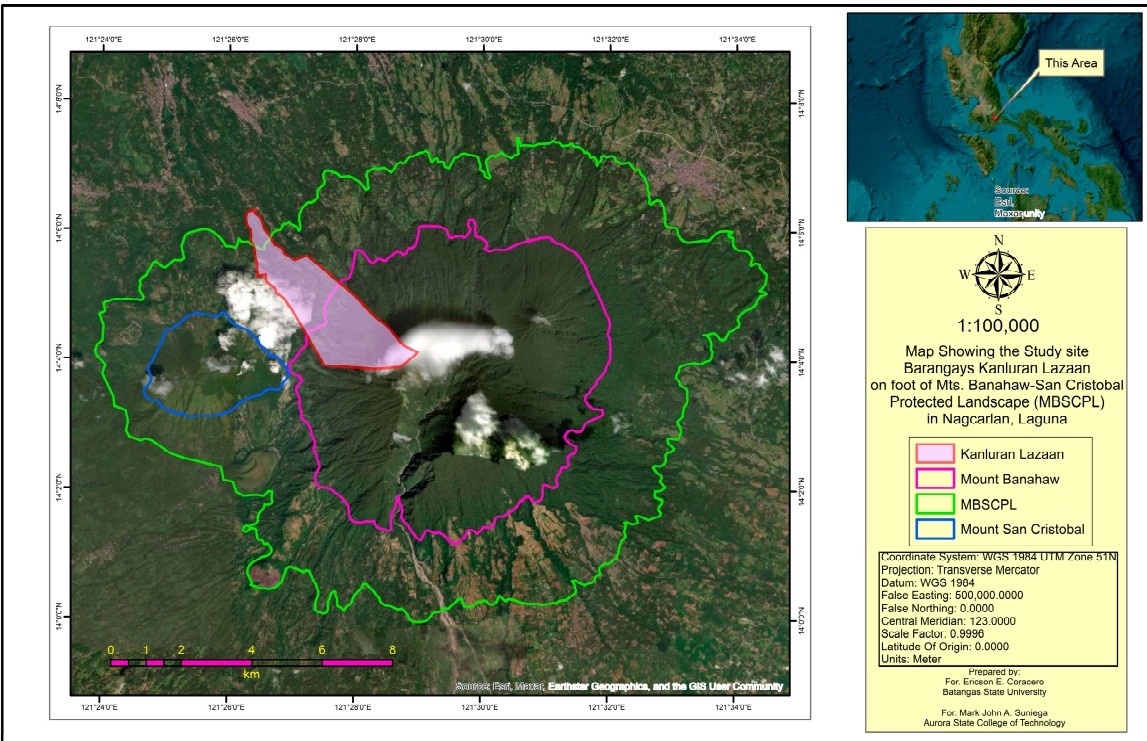

**Figure 1.** Location map of the study site.

## 2.2. Survey and Mapping of Swietenia macrophylla King

All individuals of *S. macrophylla* were inventoried and recorded. Data, such as diameter, geographical coordinates, tree height, and other notable observations (e.g., fruiting, flowering), were also recorded for each big leaf mahogany individual. Diameter was obtained at breast height (1.3 m above ground). Geographical coordinates were measured using a handheld GPS device (Garmin GPSMAP 76CSx). Tree height was estimated using a calibrated pole. Regarding the mapping procedure, the data were encoded in Microsoft excel. The recorded coordinates were converted in the Universal Transverse Mercator (UTM) format before being fed to the ArcGIS (v. 10.8) software. Then, location points of each plant individual were plotted in the map.

## 2.3. Identification of Management Strategies

For the management strategies of various government institutions, such as the Department of Environment and Natural Resources—Protected Area Management Office (DENR-PAMO), Municipal Environment and Natural Resources Office (MENRO), Sangguniang Bayan Committee on Natural Resources and Environmental Protection, and Sangguniang Barangay of Kanluran Lazaan, exhaustive interviews were performed with the office heads to determine the detailed management strategies being employed in invasive alien plant species (with emphasis on big leaf mahogany) in the barangay, in MBSCPL, and in other areas under their jurisdiction. The interviews were unstructured to really exhaust the information from them regarding the management strategies of IAPS. Unstructured interviews are believed to produce new insights and ideas that are flexible and free from error and misinterpretations [25].

*2.4. Data Analysis and Interpretation*

2.4.1. Distribution Maps

The distribution of population was presented in the form of a map showing their size classes based on DBH. For the DBH, several papers about size classes were read by the researcher. The studies of Norghauer et al. [26] on *Swietenia macrophylla* King and Itoh et al. [27] on trees in the tropical rainforest were selected based on their agreed classification of tree size classes and specially that the first study focused on *S. macrophylla,* which is the focal species of the current undertaking (Table 1).

**Table 1.** Size classes of big-leaf mahogany.

| Class | Diameter (cm) |
|---|---|
| Seedlings | <1 |
| Saplings | $1 \leq \text{Diameter} < 5$ |
| Poles | $5 \leq \text{Diameter} < 30$ |
| Adults | $\geq 30$ |

2.4.2. Content Analysis of Management Strategies

The interview responses of the heads/representatives of the four focal government agency respondents were assessed via content analysis. We followed Hermann's method [28] to perform content analysis. The steps indicated in this literature are presented in Table 2, with its corresponding equivalent in the present study.

**Table 2.** Methods of content analysis [28] and its equivalent in the present study.

| Procedures | Equivalent in the Study |
|---|---|
| Consider the research questions | Determine the management strategies implemented by the government agencies towards *S. macrophylla* |
| Select the material | Interview transcripts |
| Decide on the nature of the content analysis | Qualitative—Determined the strategies and the degree used by the government in managing *S. macrophylla* |
| Determine the unit of analysis and coding | The focus were the verbs or action words (management practice) and the adverbs (degree/frequency) to describe that verb |
| Contextualize the information | The information gathered were contextualized in a manner which looks at the management strategies employed and the intensity level of practicing them |

## 3. Results and Discussion

*3.1. S. macrophylla and Its Spatial Distribution*

A total of 1591 individuals of Mahogany plants were recorded (Table 3). Seedlings had the highest individual count with 835 (52.48%), while adults had the least with 40 individuals (2.51%). These individuals were found only in two types of locations, roadside and in mixed land use areas (anthurium, vegetable, and agroforestry farms). Only 25 (1.57%) plants were along the side of the road while 1566 (98.42%) were in the mixed land use areas.

**Table 3.** Size classes and abundance of big-leaf mahogany in the site.

| Class | Diameter (cm) | Abundance | | |
| | | Mix Land Use | Roadside | Total |
|---|---|---|---|---|
| Seedlings | <1 | 822 | 13 | 835 |
| Saplings | 1 ≤ Diameter < 5 | 83 | 6 | 89 |
| Poles | 5 ≤ Diameter < 30 | 622 | 5 | 627 |
| Adults | ≥30 | 39 | 1 | 40 |
| Total | | **1566** | **25** | **1591** |

The spatial distribution of the *S. macrophylla* population in the study site in relation to the protected area is primarily found in the buffer zones of the MBSCPL. Fortunately, no individuals of the species were found thriving beyond the buffer zone towards Mt. Banahaw de Nagcarlan. However, this close location of existence of the species still poses threats to the protected area and its native natural resources.

Regarding elevation, the populations were found along elevations of 490 to 570 masl. This elevation range satisfies the current known distribution patterns of big leaf mahogany in terms of elevation, as reports says that it can be found in areas with elevation of 0 to 1500 masl, aside from its ability to tolerate a very wide range of soils and environmental conditions [29]. Since *S. macrophylla* was introduced under unique conditions, it is impossible to compare its degree of invasiveness across elevations. It should be mentioned that the Mahogany found in the locations with the greatest concentration of people was probably planted on purpose by humans. There may have been a human-driven factor in the higher density of *S. macrophylla* in those specific regions. Because of this, comparing the degree of invasiveness based purely on elevation may not give a true reflection of the species' capacity for invasiveness in other habitats or its natural range.

Direct comparisons of the level of invasiveness across different available environmental conditions allow us to deduce that big leaf mahogany exhibits invasive qualities in the area. This conclusion is backed by the finding that six mother *S. macrophylla* trees were able to generate at least 627 offspring in a very small area of mixed land use, measuring around 500 square meters, during a period of 25 years. Big leaf mahogany has the capacity to spread and establish itself outside of its original planting places due to its prolific reproduction and quick population increase. This implies that the species has invasive traits and emphasizes the significance of tracking and controlling its spread to avoid ecological effects on indigenous ecosystems.

The distribution map of *S. macrophylla* population in Kanluran Lazaan revealed high density, especially in one area, as seen in Figures 2–6. This specific area was where the existing pioneer Mahogany trees were located. According to some key informants, big leaf mahogany was introduced in the barangay in 1998 by the Department of Forestry and Natural Resources to be planted along the roadsides. One of the barangay officials during that time planted six individuals in his land around 10 m away from the road. The individuals planted in the roadsides were already cut while six individuals planted by the barangay official were left. As observed, these six individuals became mother trees and were able to produce 627 Mahogany plants within 25 years of which majority are seedlings (~350 individuals). It was also said that some of the individuals were already transplanted to other locations. This number of individuals may be brought by the fact that a big leaf mahogany fruit can bear around 62 seeds and may be dispersed 20 to 40 m from the mother tree [7]. Also, according to the key informants, a *S. macrophylla* tree can bear fruits even before it reaches 10 years. This is relatively earlier than what previous studies have mentioned that the tree can bear fruit as early as 12 years [30]. Thus, contributing to the fruiting phenology information about the species.

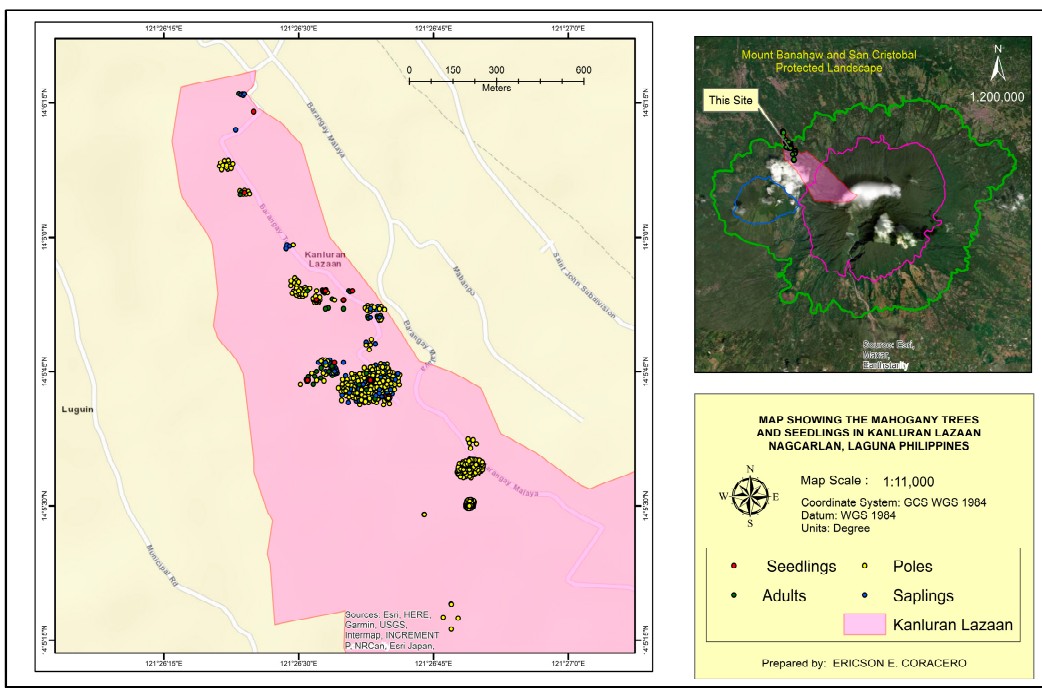

**Figure 2.** Spatial distribution map of all size classes of *S. macrophylla* populations.

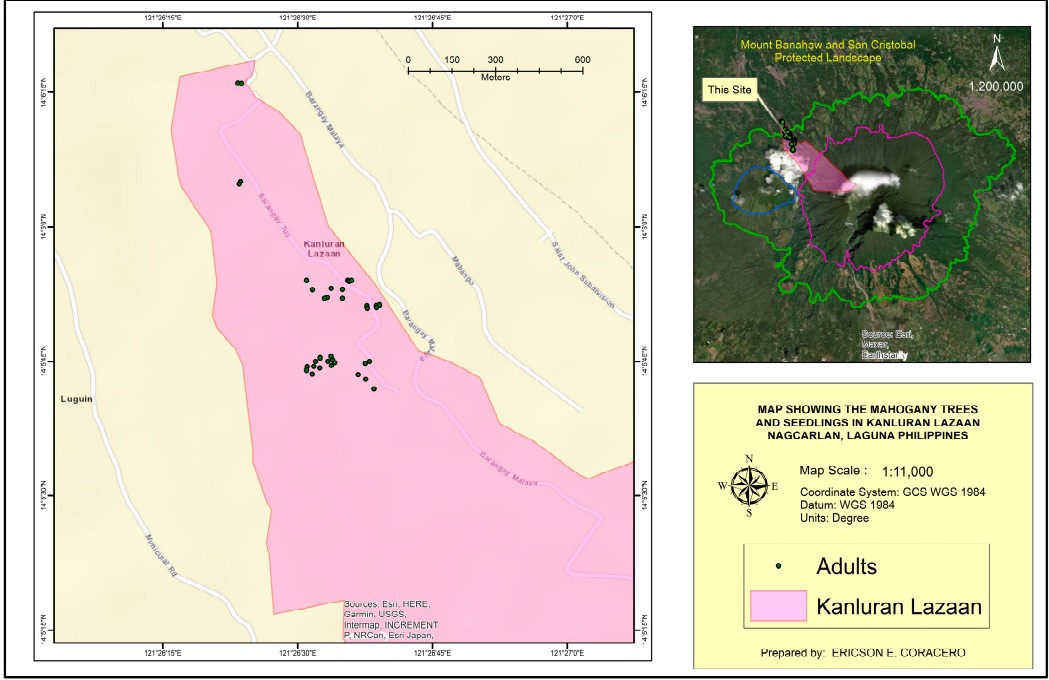

**Figure 3.** Spatial distribution map of *S. macrophylla* adult populations.

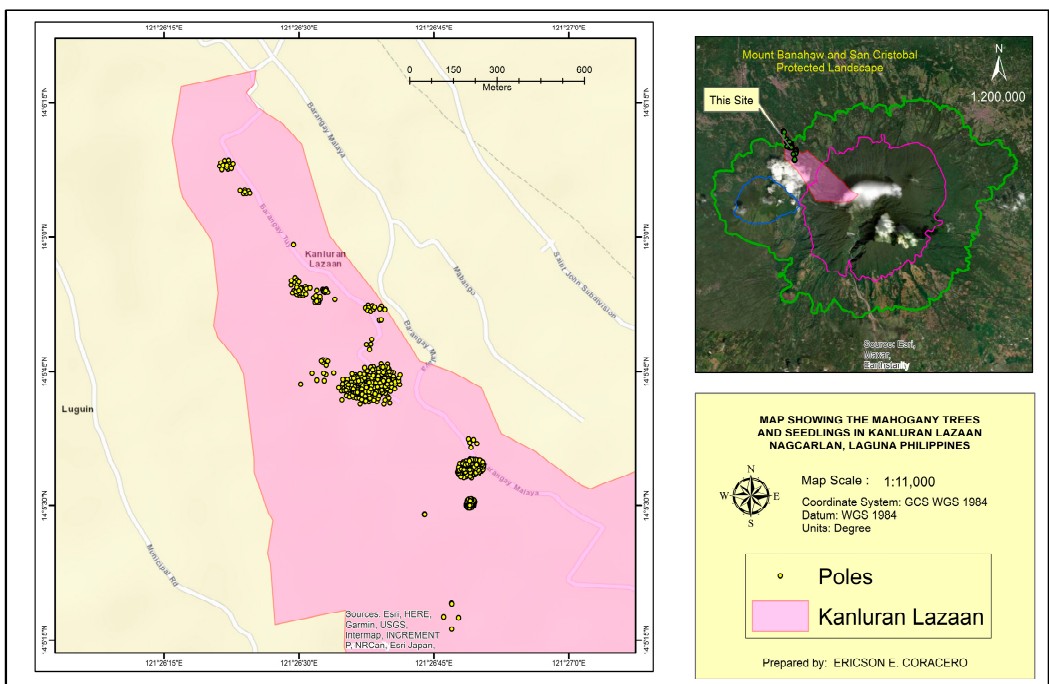

**Figure 4.** Spatial distribution map of *S. macrophylla* pole populations.

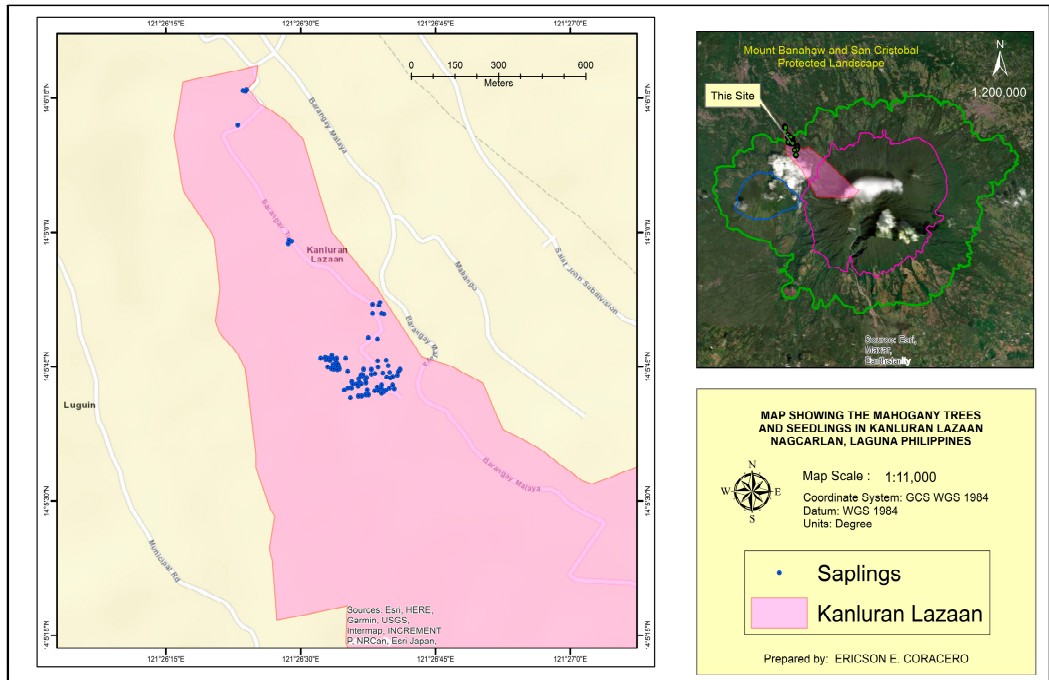

**Figure 5.** Spatial distribution map of *S. macrophylla* sapling populations.

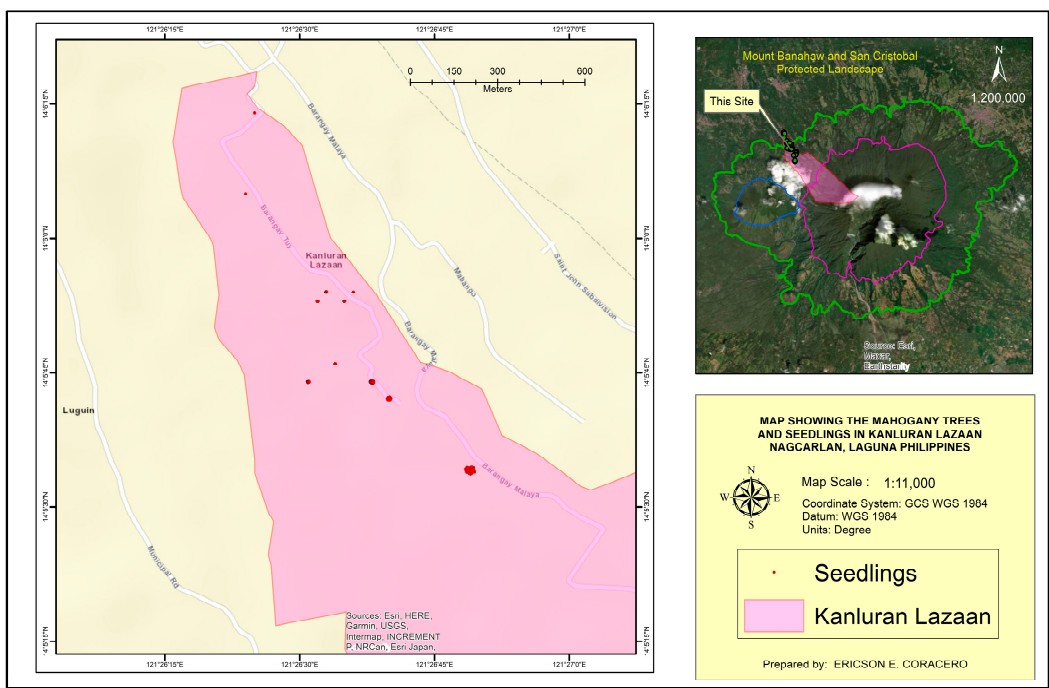

**Figure 6.** Spatial distribution map of *S. macrophylla* seedling populations.

*3.2. Existing Management Strategies of Invasive Alien Plant Species*

The interview for the management of invasive alien plant species was carried out with the head of the barangay officials of Kanluran Lazaan, head of the Protected Area Management Board (PAMB) of MBSCPL, MENR officer of Nagcarlan, and representative of the Sangguniang Bayan (SB) Committee on Natural Resources head and Environmental Protection (CNREP). Unfortunately, only the MENR officer and the SB head of CNREP were implementing strategies to manage invasive alien species.

The head of SB-CNREP mentioned that their strategies to manage invasive alien species include delivering talks about the importance of native plants and the harm of introduced and invasive plants in the ecosystem. This educational initiative is given to public and private schools, to the officials of local government units, local communities, and other stakeholders in the municipality that asks for environmental talks and lectures. Also, they aid in providing seedlings of native plants to those organizations that wish to conduct tree planting activities. The SB-CNREP worked hand-in-hand with the MENR Office with regard to the promotion and provision of native species to clienteles which include different organizations and stakeholders, such as schools, communities, and private corporations, among others. The committee also facilitates tree planting activities in Landing Point, an ecotourism site that is within the MBSCPL, wherein native plants are planted. This greening project serves both an educational and ecological purpose, as it enhances the natural ecosystem, as well as providing opportunities to learn about native plants.

The MENR officer also had somewhat similar strategies in managing invasive and introduced species. According to the MENR Officer, when she or her staff are invited to deliver talks at environmental events and seminars, they make sure to tackle this topic, especially about Mahogany being an allelopathic and a very aggressive plant that can suppress the growth of native plants. Also, they assist clients in requesting seedlings of native trees from the DENR as part of their advocacy to promote native forest and fruit trees rather than introduced and invasive plants. These species include Philippine native and/or endemic forest and fruit trees, such as *Pterocarpus indicus* Willd., *Nephelium lappaceum* L., *Artocarpus blancoi* (Elmer) Merr., *Lansium domesticum* Corrêa, *Ficus pseudopalma* Blanco, and *Lagerstroemia speciosa* (L.) Pers., among others.

Based on the strategies implemented by the government, they are on the right track. The involvement of the government highlights the significance of local governance in addressing environmental concerns. Their initiatives, including delivering talks and facilitating tree planting activities, showcase a proactive approach to raising awareness and promoting native plant species, which is acknowledged as an effective strategy to address not only IAPS but other environmental issues [31]. However, these strategies alone cannot solve the issue. The implementation of the EDRR (early detection of and rapid response) to invasive species is also an ideal action to minimize the impact of IAPS [32]. Furthermore, the municipality may opt to prioritize policy establishment and enforcement, as well as monitor and assess the situation of IAPS, which are crucial to addressing biological invasions [33,34].

Though there was involvement from MENRO and SB-CNREP, it is important to address the absence of involvement from other key stakeholders, such as sangguniang barangay and PAMB, to maximize the effectiveness of the efforts being exerted to address IAPS, which benefit from collaborative approaches [35]. In the case of this study, since the barangay officials did not have any strategies, it may have an impact on the community's knowledge and perception of IAPS.

## 4. Conclusions

This study formally confirmed the presence and rapid proliferation of *S. macrophylla* along the edges of Mt. Banahaw de Nagcarlan in Kanluran Lazaan, which highlights its invasive potential. The study found a total of 1591 individuals of *S. macrophylla* primarily distributed in mixed land use areas and roadside locations, with a concentration in a specific area where mother trees were originally planted, thus posing significant threats to the MBSCPL and calling for an urgent need to implement effective management strategies. While local government units exert efforts to raise awareness and promote native plants (e.g., *Pterocarpus indicus* Willd., *Nephelium lappaceum* L., *Artocarpus blancoi* (Elmer) Merr., *Lansium domesticum* Corrêa, *Ficus pseudopalma* Blanco, and *Lagerstroemia speciosa* (L.) Pers.), it is crucial to foster a stronger collaboration among key stakeholders to develop and enforce comprehensive management strategies in connection with the presence of IAPS. The monitoring and prevention of invasive species should continue. In conclusion, the present study emphasizes the urgent need to address the invasive nature of *S. macrophylla* in the area. Documenting its population distribution is a crucial step towards informed decision-making and planning of effective management strategies to secure the conservation and protection of the ecosystems in the area holding unique biodiversity resources.

**Funding:** This research received no external funding.

**Institutional Review Board Statement:** Not applicable.

**Data Availability Statement:** Not applicable.

**Acknowledgments:** The author is sincerely thankful to the participation of the government agencies (PAMB of MBSCPL, Sangguniang Barangay of Kanluran Lazaan, MENRO, and Municipal Government of Nagcarlan, Laguna) in the interview, helping the completion of the study. Sincere gratitude is also expressed to the author's family, partner (Jasent Gonzales Abadayan), and students (Divine, Princess Nicole, Christian, Maria Juana, Prince, Darwin, Kenneth, Loi, Cris Jude) for giving full support in the conduct of the study. Mark John Suniega is also acknowledged for the help in creating the maps. Finally, the author is thankful to his mentors RB Gallego, Michelle Resueño, Mary Jane Aragon-Marigmen, Maria Cristina Cañada, Afed Daiwey, and Jay Amon for their guidance.

**Conflicts of Interest:** The author declares no conflict of interest.

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
