# Peer review of "Distribution and Management of the Invasive Swietenia macrophylla King (Meliaceae) at the Foot of a Protected Area in Luzon Island, Philippines"

_2673-5636, doi:10.3390/jzbg4030045_

Round 1

Reviewer 1 Report

1. The word "management" should be removed, as it is different from the topic of determining the strategy about the management of the invasive species in the title.

2. Litteratür [5] not found in Manuscript.

3. References should be reviewed and written.

In addition (questions and answers below)

1. What is the main question addressed by the research?
2. Do you consider the topic original or relevant in the field? Does it
address a specific gap in the field?
3. What does it add to the subject area compared with other published
material?
4. What specific improvements should the authors consider regarding the
methodology? What further controls should be considered?
5. Are the conclusions consistent with the evidence and arguments presented
and do they address the main question posed?
6. Are the references appropriate?
7. Please include any additional comments on the tables and figures.

1.Distribution and mapping of the 1st invasive species (Swietenia macrophylla King (Meliaceae) 2. The issue is similar to invasive weeds, which is my topic. 3. In other similar studies, mapping of the invasive species is the first step and includes different applications in control and management. 4. The author's future work on the subject should include suggestions and practices on the management and control of the invasive species. Addressing the first step of the 5th main question is an important study to begin with. 6. References are appropriate. 7. Tables and figures are a start in understanding the issue and threat.     

    Monitoring and prevention of invasive species should continue.

Reviewer 2 Report

The author presented a very interesting and technically sound research article entitled " Distribution and Management of the Invasive Swietenia macrophylla King (Meliaceae) at the foot of a protected area in Luzon Island, Philippines."

Following are the comments and suggestions for the above manuscript:

Line 41: Add the family name.

Line 49: Reference seems irrelevant.

Line 62: Correct as MBSCPL

Line 74-77: Need rephrasing and suggested to write after line 68.

Section 2.

Line 80: the time period mentioned is seems lesser for observation and better to mention weather data during the study period.

Line 83-84: Repetition regarding MBSCPL and double check with previously reported information in the same manuscript.

Line 89-90: Why the temperature and other data only for the month of Feb? While the work started in March 2023.

Line 92: (Figure 1) Font size should be visible.

Section 2.2 Line 95-101: Suggested to explain inventoried data with respect to the species mentioned.

Moreover, better to add summary about number of individuals and locations (from respective organizations).

Line 109: by describing other IAPS's of the area would be more appealing and to compare with the targeted species.

Line 120: Italicize the Scientific name.

Section 2.4.2

Line 125: Describe number of responses/interviewed persons.

Section 3.

The author divided this section into two further sub-sections (Spatial distribution of the species & its management strategies).

Here only one table describes the results, whereas the Figures exhibited is much better with respect to spatial distribution. 

The results of table No. 3 should be further divided according to the size classes and mentioned zones (including Buffer Zones) or sampling sites of the study area (Line 141).

Line 166: Mentioned Fig 5 to 9, but other figures (No. 7 to 9) missing in the manuscript.

Lione172-173: Better to add Soil & Ecological information of the study site (if any).

Line 177: It simply explains bearing of fruits by key informants, but no scientific information in support of that or any finding by the author other than interview(s). It needs little bit elaboration.

Page No. 6 to 8: Figure numbers should be from Figure2 to Figure 6, hence need correction.

Line 194-196: Number of bodies/organizations, number of individuals interviewed.

Line 200-204: Suggested to describe by converting the text with some data with respect to interventions/strategies by SB-CNREP and or other organizations in the study area about IAPS.

Line 210-212: need to describe about some key native species (Forest & Fruits) of the study area by the DENR.

line 218-219: Suggest EDRR guidelines for assessment of species.

Section 4: Missing or needed to correct the respective section.

Section 5.

Line 235-236: Suggest potential Native species.

Line 238: Correction needed (IAPS).
